# Grove MoE: Towards Efficient and Superior MoE LLMs with Adjugate Experts

## Abstract

The Mixture of Experts (MoE) architecture is a cornerstone of modern state-of-the-art (SOTA) large language models (LLMs). MoE models facilitate scalability by enabling sparse parameter activation. However, traditional MoE architecture uses homogeneous experts of a uniform size, activating a fixed number of parameters irrespective of input complexity and thus limiting computational efficiency. To overcome this limitation, we introduce Grove MoE, a novel architecture incorporating experts of varying sizes, inspired by the heterogeneous big.LITTLE CPU architecture. This architecture features novel adjugate experts with a dynamic activation mechanism, enabling model capacity expansion while maintaining manageable computational overhead. Building on this architecture, we present GroveMoE-Base and GroveMoE-Inst, 33B-parameter LLMs developed by applying an upcycling strategy to the Qwen3-30B-A3B-Base model during mid-training and post-training. GroveMoE models dynamically activate 3.14-3.28B parameters based on token complexity and achieve performance comparable to SOTA open-source models of similar or even larger size.

## 1 Introduction

Recent advancements in Large Language Models (LLMs) have spurred the adoption of the Mixture of Experts (MoE) architecture (Jiang et al., 2024; Yang et al., 2025; Liu et al., 2024; Google DeepMind, 2025; Huo et al., 2025) considering its great model capacity. The MoE model operates by dynamically routing input tokens to a relevant subset of multiple experts, enhancing computational efficiency and scalability.

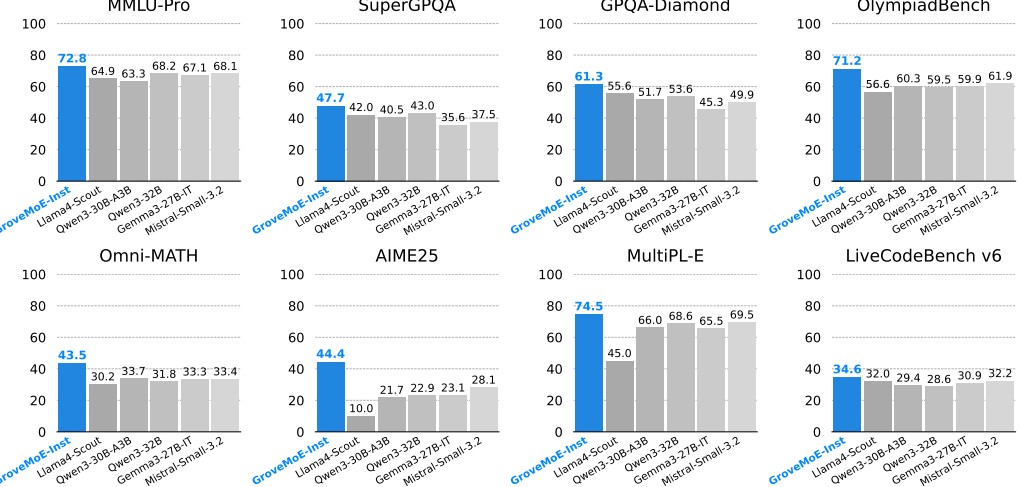

Figure 1: The performance of GroveMoE-Inst[‡] and its counterparts. GroveMoE-Inst achieves performance comparable to open-source SOTA LLMs of similar or even larger scales.

However, a key limitation of conventional MoE models is their reliance on homogeneous experts, which activates a fixed number of parameters irrespective of input token complexity. Given that token complexity varies, computational resources should ideally be allocated dynamically with more resources for complex tokens and fewer for simple ones (Huang et al., 2024). The current rigidity precludes such fine-grained control over computation. Inspiration from the big.LITTLE CPU architecture (Greenhalgh, 2011), which uses heterogeneous cores to manage computational load efficiently, we employ experts of varying sizes within MoE models could enable dynamic resource allocation based on computational demand.

Drawing inspiration from the structure of trees, we introduce Grove MoE, a novel architecture that leverages parallel adjugate experts to expand model capacity efficiently. Grove reflects our architectural design, where experts are organized into disjoint groups. Similar to how branches share a common trunk within a tree cluster, experts within a Grove group share a single adjugate expert. If multiple activated experts belong to the same group, their shared adjugate expert is computed only once before being added to each expert's output. This mechanism enables a form of dynamic computation allocation, allowing Grove MoE to expand model capacity with high computational efficiency. Crucially, the Grove MoE architecture is compatible with existing routing mechanisms, eliminating the need for complex, manually designed routing strategies (Huang et al., 2024; Wang et al., 2024b) to manage expert activation counts. Moreover, we apply a loss-free expert loading balance strategy during mid-training (Liu et al., 2024; Su, 2025).

Furthermore, the upcycling strategy (Komatsuzaki et al., 2023; Nakamura et al., 2025) has been widely used for upcycling dense models, which offer larger model capacity. Recent studies (Baykal et al., 2023; Wu et al., 2024; Chen et al., 2025) have also shown that expanding model capabilities through parallel computation, especially by reusing existing weights, is an effective strategy comparable to direct parameter scaling. Consequently, we develop our GroveMoE architecture based on pre-trained MoE models through mid-training (Meta-AI, 2025; Yang et al., 2025) and post-training stages. We summarize our main contributions as follows:

- We introduce the Grove MoE architecture, which features a new mechanism of dynamic computation allocation, allowing for parameter expansion while maintaining manageable computational costs.
- We develop GroveMoE-Base and GroveMoE-Inst, open-source models that dynamically activate 3.14-3.28B parameters, upcycled based on Qwen3-30B-A3B-Base through mid-training and post-training.
- We conduct empirical evaluations showing that our GroveMoE models achieve excellent performance comparable to open-source SOTA LLMs of similar or even larger scales.

## 2 RELATED WORKS

**big.LITTLE Architecture.** The big.LITTLE CPU architecture (Greenhalgh, 2011) offers a compelling model for computational efficiency by integrating high-performance big cores and energy-efficient LITTLE cores within a single processor, dynamically routing tasks to the appropriate core type. In contrast, traditional MoE architectures (Yang et al., 2025; Liu et al., 2024) typically employ homogeneous experts of uniform size, analogous to a processor with only one type of core, which leads to suboptimal efficiency. Drawing inspiration from the big.LITTLE architecture, we propose an MoE architecture where experts vary in computational capacity and the experts are dynamically activated. In this paper, we introduce the Grove MoE architecture, which materializes this concept through novel adjugate experts and a dynamic activation mechanism.

**MoE Architecture with Dynamic Activation.** Prior research has explored dynamic activation of expert counts in MoE models to mitigate the ineffectiveness of fixed top-$k$ routing for modeling targets of different complexity. Naive approaches (Jin et al., 2024; Zeng et al., 2024) indirectly vary the active expert count by including blank or constant experts in the routing pool. DynMoE (Guo et al., 2024) enables a top-any gating mechanism to choose any number of experts. ReMoE (Wang et al., 2024b) activates experts with positive scores via the ReLU function. Both DynMoE and ReMoE face the challenge of needing explicit mechanisms to manage the upper bound on the number of activated experts so as to avoid potential high computation overhead. Moreover, their relatively complex routing strategies are incompatible with the well-established top-$k$ routing mechanisms, which brings potential issues in practice. In contrast, our GroveMoE layer intrinsically achieves

dynamic activation by assigning adjugate experts to separate groups. This approach guarantees a controllable activation count and thus manageable computation overhead. It requires no specialized router modifications, and the excellent compatibility makes it widely applicable. We provide more comparisons in Appendix A.1.

**Upcycling Strategy.** The performance of LLMs is intrinsically related to the model capacity. Various upcycling methods (Komatsuzaki et al., 2023; Nakamura et al., 2025) have been proposed to upcycle dense models, leveraging knowledge from pre-training models. This paper develops GroveMoE-Base and GroveMoE-Inst models by applying the upcycle strategy to the MoE model Qwen3-30B-A3B-Base (Yang et al., 2025).

## 3 ARCHITECTURE

### 3.1 TRADITIONAL MoE LAYER

An MoE layer comprises $n$ experts $\{E_i\}_{i=1}^n$ and a router $R$. Let $\boldsymbol{x} \in \mathbb{R}^d$ represent the feature of an input token, where $d$ denotes the feature dimension. The routing scores are calculated as $\boldsymbol{\rho} = R(\boldsymbol{x}) \in \mathbb{R}^n$ and the output of the $i$-th expert is $\boldsymbol{e}_i = E_i(\boldsymbol{x}) \in \mathbb{R}^d$. The final output of the MoE layer for each token is a weighted sum of the outputs from a selected subset of experts:

$$\boldsymbol{y} = \sum_{i \in \arg \operatorname{top}k(\boldsymbol{\rho})} \rho_i \boldsymbol{e}_i, \tag{1}$$

where $\arg \operatorname{top}k(\cdot)$ selects the indices of the top $k$ routing scores.

### 3.2 GROVE MoE WITH ADJUGATE EXPERTS

Expanding model capacity during mid-training can preserve existing knowledge while providing additional resources to acquire complex skills. However, under the upcycling strategy, directly duplicating each expert's parameters would disturb the original distribution of $\boldsymbol{\rho}$. Specifically, when experts are duplicated, $k$ must be scaled proportionally by the same scale factor. However, $k$ should be controlled to avoid introducing significant activation parameters. Alternatively, extending the parameters of each expert $E_i$ maintains the original $\boldsymbol{\rho}$ distribution and offers a viable solution.

The AltUp architecture (Baykal et al., 2023) demonstrates that introducing parallel blocks can increase model capacity without incurring substantial computational overhead. Moreover, scaling parallel computation by reusing existing parameters has proven effective for enhancing model capability (Wu et al., 2024; Chen et al., 2025). Inspiredly, we introduce Grove MoE, a novel architecture that leverages parallel adjugnate experts for efficient model capacity expansion. In the Grove MoE layer, we divide $n$ experts, $\{E_i\}_{i=1}^n$, into $g$ disjoint groups, where $g$ divides $n$, and each group contains $n/g$ experts. The groups $\{G_j\}_{j=1}^g$ can be defined as follows:

$$G_j = \left\{ E_i \mid \left\lfloor \frac{i-1}{n/g} \right\rfloor + 1 = j \right\}, \text{ for } j = 1, 2, \dots, g, \tag{2}$$

where $\lfloor \cdot \rfloor$ denotes the floor function. Motivated by big.LITTLE architecture (Greenhalgh, 2011), we introduce an adjugate expert $A_j$ for each group $G_j$. Notably, the capacity of the adjugate expert $A_j$ can differ from that of the expert $E_i$. The modified output $\bar{e}_i$ is then computed as:

$$\bar{e}_i = E_i(\boldsymbol{x}) + \lambda A_j(\boldsymbol{x}), \text{ where } j = \left\lfloor \frac{i-1}{n/g} \right\rfloor + 1. \tag{3}$$

Here, $\lambda$ is the scaling factor for the adjugate expert. The final output of the Grove MoE layer is:

$$\boldsymbol{y} = \sum_{i \in \arg \operatorname{top}k(\boldsymbol{\rho})} \rho_i \bar{e}_i = \sum_{i \in \arg \operatorname{top}k(\boldsymbol{\rho})} \rho_i(E_i(\boldsymbol{x}) + \lambda A_j(\boldsymbol{x})), \text{ where } j = \left\lfloor \frac{i-1}{n/g} \right\rfloor + 1. \tag{4}$$

The key advantage of the Grove MoE architecture is dynamic computation allocation. To illustrate, consider experts $E_r$ and $E_s$ where $\left\lfloor \frac{r-1}{n/g} \right\rfloor = \left\lfloor \frac{s-1}{n/g} \right\rfloor$. According to Equation (4):

$$\rho_r \bar{e}_r + \rho_s \bar{e}_s = \rho_r(E_r(\boldsymbol{x}) + \lambda A_j(\boldsymbol{x})) + \rho_s(E_s(\boldsymbol{x}) + \lambda A_j(\boldsymbol{x}))$$

$$= \rho_r E_r(\boldsymbol{x}) + \rho_s E_s(\boldsymbol{x}) + (\rho_r + \rho_s)\lambda A_j(\boldsymbol{x}), \text{ where } j = \left\lfloor \frac{r-1}{n/g} \right\rfloor + 1. \tag{5}$$

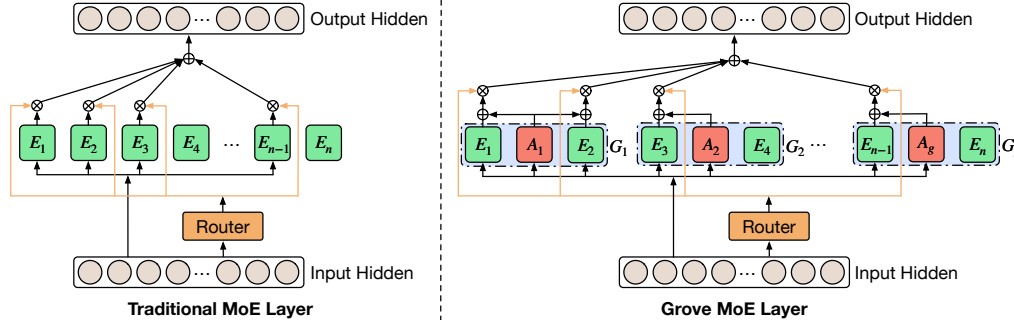

Figure 2: Comparison between the traditional MoE layer and our Grove MoE layer with dynamic computation allocation. For clarity, we configure $n/g = 2$ and $k = 4$ for the Grove MoE layer.

For the adjugate expert $A_j$, the routing weight $\rho_A = (\rho_r + \rho_s)\lambda$ should be restricted to be no more than the weight of experts $E_r$ and $E_s$, especially for the upcycling mid-training case. Generally, we restrict $\lambda \leq 1.0/(n/g) = g/n$. For instance, for a MoE model with $n$=128 experts and $g$=64 groups, we restrict $\lambda \leq 0.5$.

As shown in Figure 2, if multiple activated experts belong to the same group, their adjugate expert $A_j$ is computed only once before being added to each expert's output, scaled by the sum of their routing weights. Meanwhile, if all experts are not activated, their adjugate expert is also not computed. According to Equation (5), the number of activated adjugate expert $A_j$ ranges from $\left\lceil \frac{k}{n/g} \right\rceil$ to $k$ for each Grove MoE layer, where $\lceil \cdot \rceil$ denotes the ceiling function. This mechanism enables a form of dynamic computation allocation, allowing Grove MoE to expand model capacity with high computational efficiency.

### 3.3 EXPERTS LOADING BALANCE

In MoE models, workload imbalance among experts can induce routing collapse and reduce computational efficiency. To better balance load distribution while maintaining model performance, we employ an auxiliary-loss-free load balancing strategy (Liu et al., 2024). Specifically, we introduce a dynamic expert-wise bias term $\boldsymbol{b}$ to adjust routing scores $\boldsymbol{\rho}$. The gating mechanism in Equation (4) is therefore modified as:

$$\boldsymbol{y} = \sum_{i \in \arg \operatorname{top}k(\boldsymbol{\rho}+\boldsymbol{b})} \rho_i \bar{\boldsymbol{e}}_i, \tag{6}$$

where $\boldsymbol{b}$ is updated via sign gradient descent to minimize imbalance.

Formally, we define $\boldsymbol{F} = \mathbb{E}(\boldsymbol{f})$ as the current load distribution across experts under the bias $\boldsymbol{b}$, where $\boldsymbol{f} = [f_1, f_2, \cdots, f_n]$ and $f_i$ denotes the assignment probability for each token:

$$f_i = \begin{cases} 1/k, & i \in \arg \operatorname{top}k(\boldsymbol{\rho} + \boldsymbol{b}), \\ 0, & \text{otherwise.} \end{cases} \tag{7}$$

The uniform load distribution is $\boldsymbol{Q} = [\frac{1}{n}, \frac{1}{n}, \cdots, \frac{1}{n}]$. To optimize load balancing (Su, 2025), $\boldsymbol{b}$ is updated as:

$$\boldsymbol{b} \leftarrow \boldsymbol{b} - \alpha \frac{\boldsymbol{F} - \boldsymbol{Q}}{\sqrt{\frac{1}{n} \sum_{i=1}^{n} (F_i - Q_i)^2}}. \tag{8}$$

Specifically, Equation (8) uses RMSNorm to normalize the imbalance $(\boldsymbol{F} - \boldsymbol{Q})$ for better workload balance. To address the sensitivity of the parameter $\alpha$ in Equation (8), resulting from its coupling with the Sigmoid router (Liu et al., 2024; Su, 2025), we decouple the routing calculation. The final output $\boldsymbol{y}$ is computed as:

$$\boldsymbol{y} = \sum_{i \in \arg \operatorname{top}k(\boldsymbol{\rho}^{(\sigma)}+\boldsymbol{b})} \rho_i^{(h)} \boldsymbol{e}_i, \tag{9}$$

Table 1: Architecture exploration on different expert group settings. The highest and second-best scores are shown in bold and underlined, respectively.

| Expert Groups | - | - | $g = 64$; $h = 128$ | $g = 32$; $h = 256$ | $g = 16$; $h = 512$ |
|---|---|---|---|---|---|
| Architecture | MoE | MoE | Grove MoE | Grove MoE | Grove MoE |
| # Total Params | 30B | 30B | 33B | 33B | 33B |
| # Activated Params | 3B | 3B | 3.14B-3.28B | 3.14B-3.57B | 3.14B-4.11B |
| # Avg. Activation | 3B | 3B | 3.26B | 3.51B | 3.93B |
| # Training Tokens | 0B | 50B | 50B | 50B | 50B |
| MMLU | 81.58 | 82.56 | **82.57** | 82.12 | 80.23 |
| CMMLU | 80.63 | **86.54** | 86.47 | 85.54 | 85.57 |
| SuperGPQA | 36.10 | 35.93 | **36.22** | 36.09 | 36.12 |
| GSM8K | 89.39 | 89.54 | 90.07 | **90.83** | 90.67 |
| MATH | 59.75 | 65.90 | 66.52 | 66.60 | **66.64** |
| GPQA-Diamond | 39.39 | 38.38 | 41.41 | 39.39 | **44.95** |
| HumanEval+ | 83.54 | 83.54 | **85.37** | 84.75 | 84.15 |
| MBPP+ | 71.96 | 74.34 | **75.66** | 75.13 | 74.07 |
| Average | 67.79 | 69.59 | **70.54** | 70.06 | 70.30 |

Table 2: Architecture exploration on different scaling factors. The highest and second-best scores are shown in bold and underlined, respectively.

| Scaling Factor | - | - | $\lambda = 0.20$ | $\lambda = 0.10$ | $\lambda = 0.05$ |
|---|---|---|---|---|---|
| Architecture | MoE | MoE | Grove MoE | Grove MoE | Grove MoE |
| # Total Params | 30B | 30B | 33B | 33B | 33B |
| # Training Tokens | 0B | 50B | 50B | 50B | 50B |
| MMLU | 81.58 | 82.56 | 79.69 | 82.57 | **82.62** |
| CMMLU | 80.63 | 86.54 | 86.33 | 86.47 | **86.55** |
| SuperGPQA | 36.10 | 35.93 | 33.99 | 36.22 | **36.32** |
| GSM8K | 89.39 | 89.54 | 90.14 | 90.07 | **90.83** |
| MATH | 59.75 | 65.90 | **66.62** | 66.52 | 65.86 |
| GPQA-Diamond | 39.39 | 38.38 | 43.43 | 41.41 | **43.94** |
| HumanEval+ | 83.54 | 83.54 | **85.37** | **85.37** | 84.76 |
| MBPP+ | 71.96 | 74.34 | **75.93** | 75.66 | 75.13 |
| Average | 67.79 | 69.59 | 70.19 | 70.54 | **70.75** |

where $\rho^{(h)}$ is the output of a Softmax router and $\boldsymbol{\rho}^{(\sigma)}$ is the output of a Sigmoid router:

$$\rho_i^{(h)} = \frac{e^{x_i}}{\sum_{j=1}^n e^{x_j}} \text{ and } \rho_i^{(\sigma)} = \frac{1}{1 + e^{-x_i}}. \tag{10}$$

This decoupled approach enables using a constant value of $\alpha = 0.001$, as used in Liu et al. (2024).

### 3.4 REUSE OF PRE-TRAINED WEIGHTS

Building on the concept of upcycling strategy (Komatsuzaki et al., 2023; Nakamura et al., 2025; Wu et al., 2024), we leverage pre-trained weights from MoE models. During initialization of our Grove MoE architecture, each expert $E_i$ is derived from a pre-trained MoE layer. To ensure structural coherence, other components, such as the normalization and attention layers, are directly copied from the pre-trained transformer block. Additionally, the down-projection blocks of newly inserted modules $\{A_j\}_{j=1}^g$ are zero-initialized. The remaining weights in $\{A_j\}_{j=1}^g$ follow a normal distribution with a standard deviation of 0.006 (Liu et al., 2024).

## 4 MID-TRAINING

### 4.1 MID-TRAINING SETTING

The mid-training stage is designed to target specific proficiencies, such as reasoning and code generation. The corpus utilized for this stage is a diverse collection of textual and non-textual data, encompassing sources including web content, books, academic papers, mathematics, programming code, etc. In total, the high-quality corpus consists of approximately 400 billion tokens.

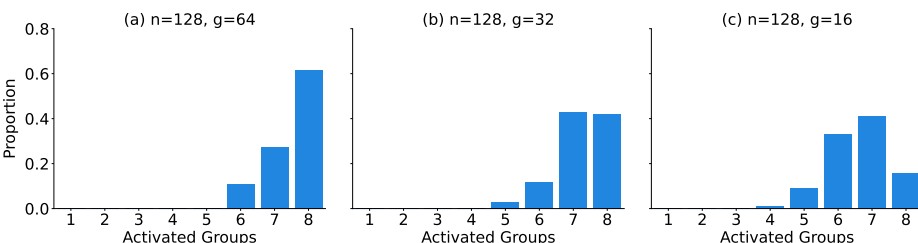

Figure 3: The group routing distribution across three configurations with different group numbers $g$. As the number of groups decreases, the average number of adjugate experts activations decreases.

Moreover, our comprehensive evaluation of the base models developed via mid-training assesses five core capabilities: general knowledge, scientific knowledge, reasoning, mathematics, and coding. The details of the evaluation benchmarks for mid-training are illustrated in Appendix B.1.

## 4.2 ARCHITECTURE EXPLORATION

We explored several key design choices for the model architecture, focusing on the configuration of the expert groups and the scaling factor in our Grove MoE architecture. The architecture of the shared parallel experts $\{A_j\}_{j=1}^g$ adopts the design established in Qwen3 MoE architecture (Yang et al., 2025). Architectural exploration is conducted using 50B tokens sampled from the mid-training dataset, with evaluations performed across diverse task types. The exploration is based on the Qwen3-30B-A3B-Base model (Yang et al., 2025), using direct mid-training without upcycling as the baseline.

**Expert Groups.** As detailed in Table 1, we evaluate the impact of expert group configurations on model performance while maintaining approximately 33B total parameters. Three configurations are compared: (1) $g = 64$, $h = 128$; (2) $g = 32$, $h = 256$; and (3) $g = 16$, $h = 512$, where $h$ denotes the intermediate dimension of $\{A_j\}_{j=1}^g$. For general knowledge, language understanding, and code generation, configuration with $g = 64$, $h = 128$ achieves optimal performance. Meanwhile, a configuration with $g = 16$, $h = 512$ yields superior results for complex mathematical reasoning and STEM tasks. Notably, three configurations outperform the baseline in terms of average performance, demonstrating the effectiveness of our Grove MoE architecture for expanding model capacity.

**Scaling Factor.** Table 2 evaluates the influence of the expert output scaling factor $\lambda$. For general knowledge and language understanding, $\lambda = 0.05$ achieves peak performance. For mathematics and STEM tasks, both $\lambda = 0.05$ and $\lambda = 0.20$ excel, while $\lambda = 0.20$ is optimal for code generation. In general, Grove MoE with a smaller scaling factor outperforms the baseline across multiple tasks.

**Group Routing Analysis.** To analyze the group routing distribution, we sample 1 million tokens from the mid-training dataset. For an LLM with $n = 128$ experts, Figure 3 illustrates the distribution across three configurations. Notably, when analyzing the routing patterns across diverse task domains, including STEM, coding, and general knowledge, the distributions remain largely consistent and similar to the overall pattern illustrated in Figure 3. With a large number of groups ($g = 64$), expert activation is broadly distributed, with most experts assigned to $7 - 8$ groups. In contrast, configurations with fewer groups ($g = 32$ and $g = 16$) exhibit highly consolidated expert activation. This consolidation directly impacts computational efficiency. With the $g = 64$ configuration, the average number of activated parameters is 3.26B, reducing computation by approximately 5%. As the number of groups decreases, the computational savings increase. For the $g = 16$ configuration, the savings reach approximately 20%.

## 4.3 HYPER-PARAMETERS

The GroveMoE-Base model is trained based on Qwen3-30B-A3B-Base (Yang et al., 2025). We employ the AdamW optimizer (Loshchilov & Hutter, 2017) with $\beta_1 = 0.9$, $\beta_2 = 0.95$, weight decay of 0.1, and gradient clipping at 1.0. Training uses a maximum sequence length of 8192 tokens, with the model trained on 400 billion tokens at a batch size of 16 million tokens. During the mid-training stage, we employ a cosine learning rate scheduler to decay the learning rate from

Table 3: Comparison among GroveMoE-Base and other strong open-source baselines. The highest and second-best scores are shown in bold and underlined, respectively.

| | Mistral-Small-3.1 Base-2503 | Gemma3-27B Base | Qwen2.5-32B Base | Qwen3-30B-A3B Base | Llama4-Scout Base | GroveMoE Base |
|---|---|---|---|---|---|---|
| Architecture | Dense | Dense | Dense | MoE | MoE | Grove MoE |
| # Total Params | 24B | 27B | 32B | 30B | 109B | 33B |
| # Activated Params | 24B | 27B | 32B | 3B | 17B | 3.14B-3.28B |
| **General Tasks** | | | | | | |
| MMLU | 81.65 | 79.89 | **83.50** | 81.58 | 79.08 | 82.86 |
| MMLU-Pro | 55.67 | 52.97 | 59.04 | **59.58** | 57.32 | 59.06 |
| CMMLU | 74.85 | 70.17 | **88.17** | 80.63 | 76.05 | 86.75 |
| SuperGPQA | 30.47 | 30.15 | 35.80 | 36.10 | 27.54 | **38.74** |
| BBH | 83.46 | 79.19 | **84.30** | 81.58 | 82.60 | 82.09 |
| C-Eval | 72.81 | 70.00 | 86.96 | 87.82 | 74.80 | **87.84** |
| **Math & STEM Tasks** | | | | | | |
| GSM8K | 85.90 | 82.71 | 90.45 | 89.39 | 86.43 | **90.83** |
| MATH | 43.90 | 49.80 | 60.42 | 59.75 | 51.34 | **64.82** |
| GPQA-Diamond | 39.90 | 36.36 | 41.41 | 39.39 | 37.54 | **41.92** |
| **Coding Tasks** | | | | | | |
| HumanEval+ | 60.98 | 57.32 | 78.05 | 83.54 | 64.63 | **85.98** |
| MBPP+ | 71.16 | 69.84 | 73.81 | 71.96 | 69.84 | **76.19** |
| MultiPL-E | 27.32 | 48.20 | 52.57 | **61.76** | 48.53 | 60.38 |
| CRUX-O | 50.38 | 60.12 | 67.88 | 67.20 | 59.54 | **70.25** |

$1 \times 10^{-4}$ to $5 \times 10^{-5}$. Consistent with our architectural analysis in Section 4.2, we configure the $\{A_j\}_{j=1}^{g}$ modules within the Grove MoE layer with group number $g = 64$, intermediate size $h = 128$, and scaling factor $\lambda = 0.05$.

## 4.4 MID-TRAINING EVALUATION

We compare our GroveMoE-Base models with leading open-source base models, including Mistral-Small-3.1-Base-2503 (Mistral AI, 2025), Gemma3-27B-Base (Gemma et al., 2025), Qwen2.5-32B-Base (Yang et al., 2024), Qwen3-30B-A3B-Base (Yang et al., 2025), and Llama4-Scout-Base (Meta-AI, 2025). All models are evaluated using the same evaluation pipeline and the widely used evaluation settings (Yang et al., 2025) to ensure fair comparison.

As depicted in Table 3, our GroveMoE-Base model, built on the Grove MoE architecture, achieves a strong balance of performance and efficiency. Our Grove MoE architecture enables operation with fewer activated parameters than dense models and achieves greater parameter efficiency than other MoE baselines. GroveMoE-Base excels at complex reasoning, surpassing all competing baselines in Math & STEM and coding benchmarks to achieve the highest average scores. In addition to these advanced reasoning capabilities, GroveMoE-Base is highly competitive in general tasks. It matches the performance of Qwen2.5-32B-Base, a dense model with a similar total parameter count, while maintaining its advantage in activation efficiency.

Notably, GroveMoE-Base is developed based on Qwen3-30B-A3B-Base. As shown in Table 3, the Grove MoE architecture facilitates an efficient expansion of model capacity with only a minor additional computational cost. The Grove MoE architecture allows the model to preserve foundational knowledge while providing dedicated resources to master new, complex skills.

## 5 POST-TRAINING

### 5.1 SUPERVISED FINE-TUNING

Following the mid-training stage, the GroveMoE-Base model undergoes supervised fine-tuning (SFT). This stage is crucially dependent on the training data. Given the scarcity and high annotation cost of human-generated data, synthetic data has become increasingly important.

Our SFT dataset is constructed from a seed set of 1–2 million instances, comprising human annotations and open-source materials. This initial dataset is then substantially expanded through a data

Table 4: Comparison among GroveMoE-Inst and other strong open-source non-reasoning baselines. The highest and second-best scores are shown in bold and underlined, respectively.

| | Mistral-Small-3.2 Instruct-2506 | Gemma3-27B IT | Qwen3-32B Non-Thinking | Qwen3-30B-A3B Non-Thinking | Llama4-Scout | GroveMoE Inst |
|---|---|---|---|---|---|---|
| Architecture | Dense | Dense | Dense | MoE | MoE | Grove MoE |
| # Total Params | 24B | 27B | 32B | 30B | 109B | 33B |
| # Activated Params | 24B | 27B | 32B | 3B | 17B | 3.14B-3.28B |
| **General Tasks** | | | | | | |
| MMLU | 80.29 | 75.97 | 82.93 | 80.12 | 81.88 | **88.04** |
| MMLU-Pro | 68.11 | 67.10 | 68.25 | 63.30 | 64.92 | **72.78** |
| CMMLU | 74.02 | 65.82 | 84.63 | 83.13 | 76.12 | **86.66** |
| SuperGPQA | 37.53 | 35.63 | 43.04 | 40.50 | 42.02 | **47.69** |
| BBH | 85.51 | 85.79 | 85.45 | 82.55 | 77.37 | **88.42** |
| DROP | 86.02 | 87.81 | 84.02 | 86.38 | 88.26 | **88.84** |
| C-Eval | 72.01 | 67.31 | 87.53 | 85.95 | 74.69 | **87.60** |
| AGIEval | 58.24 | 53.63 | 63.64 | 65.27 | 62.31 | **82.19** |
| **Alignment Tasks** | | | | | | |
| IFEval | 82.52 | 86.14 | 85.27 | 84.55 | 85.57 | **86.54** |
| Arena-Hard | 83.87 | 89.38 | 90.49 | 88.33 | 73.49 | **92.01** |
| **Math & STEM Tasks** | | | | | | |
| MATH | 84.18 | 85.82 | 85.26 | 84.68 | 81.46 | **90.56** |
| MATH-500 | 86.50 | 87.80 | 87.40 | 88.70 | 82.60 | **94.60** |
| Omni-MATH | 33.40 | 33.30 | 31.80 | 33.70 | 25.78 | **43.50** |
| AIME24 | 36.88 | 29.58 | 27.71 | 28.33 | 28.60 | **54.58** |
| AIME25 | 28.12 | 23.12 | 22.92 | 21.67 | 10.00 | **44.38** |
| GPQA-Diamond | 49.94 | 45.33 | 53.60 | 51.71 | 55.56 | **61.30** |
| OlympiadBench | 61.89 | 59.85 | 59.52 | 60.26 | 56.11 | **71.22** |
| **Coding & Agent Tasks** | | | | | | |
| HumanEval+ | 81.94 | 78.81 | 82.93 | 84.15 | 79.88 | **90.24** |
| MBPP+ | 73.54 | 73.83 | 72.75 | 75.16 | 70.37 | **78.31** |
| MultiPL-E | 69.49 | 65.50 | 68.62 | 66.04 | 45.00 | **74.53** |
| LiveCodeBench v5 | 25.90 | 26.75 | 31.44 | 28.89 | 25.45 | **33.38** |
| LiveCodeBench v6 | 32.25 | 30.86 | 28.57 | 29.43 | 32.04 | **34.60** |
| BFCL v3 (Live) | **78.21** | 75.31 | 75.09 | 73.69 | 45.41 | 76.11 |

synthesis pipeline. Our data synthesis process begins by generating novel prompts using methods inspired by Magpie-style approaches (Xu et al., 2024) and OSS-Instruct (Wei et al., 2023). We then apply rejection sampling (Grattafiori et al., 2024) to produce candidate responses using various LLMs (Yang et al., 2025; Google DeepMind, 2025; Hurst et al., 2024). To ensure high data quality, we employ a multi-stage filtering process. Initially, rule-based filters are applied to reasoning-intensive data, such as code, mathematics, and logic problems. Subsequently, all data types undergo a final assessment by an LLM-based evaluator, which uses a detailed rubric to verify the response quality and relevance. This rigorous data curation process yields a robust dataset for SFT.

Moreover, to evaluate the quality of instruction-tuned models, we evaluate LLMs on a series of post-training benchmarks, the details of which are illustrated in Appendix B.2.

## 5.2 HYPER-PARAMETERS

The post-training stage uses the AdamW optimizer (Loshchilov & Hutter, 2017), with $\beta_1 = 0.9$, $\beta_2 = 0.95$, weight decay of 0.1, and gradient clipping at 1.0. During the post-training stage, we employ a cosine learning rate scheduler with a learning rate of $5 \times 10^{-6}$ that gradually decays to a minimum of $1 \times 10^{-6}$.

## 5.3 POST-TRAINING EVALUATION

We compare our GroveMoE-Inst with leading open-source LLMs, including Mistral-Small-3.2-Instruct-2506 (Mistral AI, 2025), Gemma3-27B-IT (Gemma et al., 2025), Qwen3-32B (Yang et al., 2025), Qwen3-30B-A3B (Yang et al., 2025), and Llama4-Scout (Meta-AI, 2025).

As shown in Table 4, GroveMoE-Inst establishes excellent performance across a comprehensive set of benchmarks, maintaining high parameter efficiency. In general and alignment tasks, the

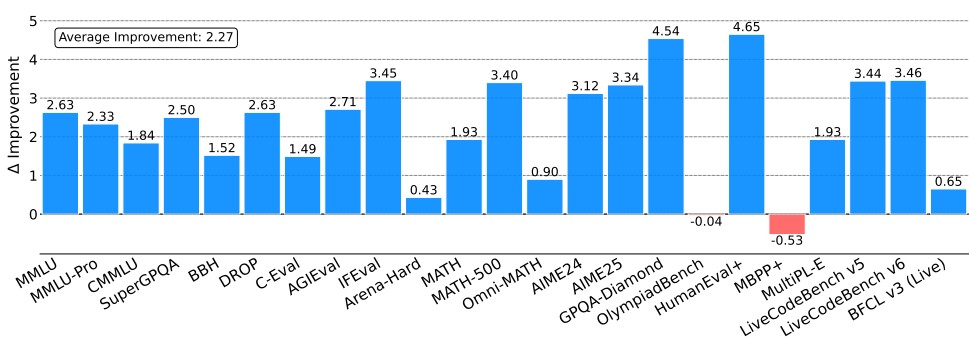

Figure 4: Evaluation results of SFT on various benchmarks. Δ indicates the performance improvement of SFT trained with GroveMoE-Base over Qwen3-30B-A3B-Base.

model consistently outperforms its counterparts, securing the highest scores on all benchmarks. The superiority of our GroveMoE-Inst is particularly pronounced in mathematics and STEM, where GroveMoE-Inst ranks first across all listed benchmarks, highlighting its powerful reasoning capabilities. Furthermore, GroveMoE-Inst demonstrates exceptional performance in coding and agent-based tasks. It surpasses other baselines on most coding & agent benchmarks, which underscores its advanced skills in code generation and problem-solving.

### 5.4 EFFECTIVENESS OF GROVEMOE-BASE

To evaluate the effectiveness of our GroveMoE-Base model, we applied the same post-training strategy to Qwen3-30B-A3B-Base (Yang et al., 2025) for a direct comparison. We also provide additional ablation study in Appendix C.2 for fair comparison under the same training corpus.

As shown in Figure 4, the instruction-tuned model derived from GroveMoE-Base consistently outperforms its Qwen3-30B-A3B-Base counterpart across the vast majority of tasks, highlighting its strong potential as a foundation model. Specifically, the GroveMoE-Base model achieves higher scores on nearly all general and alignment benchmarks. The advantage is further pronounced in specialized domains, where it significantly outperforms the Qwen model on most mathematics and STEM benchmarks. Furthermore, its superiority extends to code generation and agent-based tasks, where it secures stronger results on key benchmarks.

In summary, these results demonstrate that GroveMoE-Base is a more powerful foundation model. Its larger model capacity enables fine-tuned derivatives to achieve superior performance across a wide spectrum of domains, including general knowledge, mathematics, and coding.

## 6 CONCLUSION

This paper introduces GroveMoE models, efficient and open-source LLMs built upon the Grove MoE architecture, which incorporates a novel mechanism for dynamic computation allocation. The Grove MoE architecture improves computational efficiency by dividing experts into groups, each with an adjugate expert. This design ensures that shared computations are performed only once per group, even when multiple experts are activated. GroveMoE-Base and GroveMoE-Inst are 33B-parameter models developed based on the Qwen3-30B-A3B-Base model using our Grove MoE architecture during the mid-training and post-training stage. It dynamically activates 3.14–3.28B parameters per token. Empirical evaluations demonstrate that our GroveMoE models, including Base and Inst models, achieve performance comparable to SOTA open-source models of similar or even larger sizes, thereby validating the effectiveness of the Grove MoE architecture.

### DECLARATION OF LLM USAGE

The usage of LLMs is strictly limited to aid and polish the paper writing.

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

## A MoE Architecture

### A.1 Dynamic Activation

Existing dynamic activation methods in MoE models aim to solve the limitations of fixed top-$k$ routing. However, current approaches either indirectly vary expert counts using blank experts (Jin et al., 2024; Zeng et al., 2024) or employ complex routing strategies (Guo et al., 2024; Wang et al., 2024b) that suffer from uncontrollable computational overhead and incompatibility with standard top-k mechanisms. In contrast, as shown in Table 5, our GroveMoE intrinsically achieves dynamic activation through expert grouping. This design ensures that both the number of active experts and the computational cost are controllable. Furthermore, it requires no router modifications, making it highly compatible and easy to implement.

Table 5: Comparison among MoE Architectures with Dynamic Activation.

| Scheme | Dynamism | Computational Control | Compatibility | Parameter Efficiency |
|---|---|---|---|---|
| **Fixed Top-$k$** (Jin et al., 2024; Zeng et al., 2024) | ✗ Rigid | ✓ Strict | ✓ High | ✗ Low |
| **Dynamic Top-$k$** (Guo et al., 2024; Wang et al., 2024b) | ✓ Flexible | ✗ Explicit Constraints | ✗ Redesign of the Router | ✗ Low |
| **GroveMoE (Ours)** | ✓ Flexible | ✓ Implicit Grouping | ✓ Compatible with Top-$k$ | ✓ Shared Adjudication |

### A.2 Heterogeneous Experts

The MoE layer with heterogeneous experts proposes that experts should vary in size to possess diverse capacities (Wang et al., 2025; Sun et al., 2024). This heterogeneity allows the model to route input tokens to the expert with appropriate capability, thereby more effectively handling the varying complexity of the data and enabling greater expert specialization. In this research, our utilization of adjugate experts with varying sizes serves as a deliberate design choice to control the granularity of the activated parameter counts. This structure ensures that the step size for dynamic activation is neither excessively large nor too small, enabling finer control over the activated model capacity.

## B Evaluation Benchmarks

### B.1 Evaluation Benchmarks for Mid-Training

Our comprehensive evaluation of the base models assesses five core capabilities: general knowledge, scientific knowledge, reasoning, mathematics, and coding. The evaluation is conducted using 13 distinct benchmarks:

- **General Tasks:** MMLU (Hendrycks et al., 2020)(5-shot), MMLU-Pro (Wang et al., 2024a)(5-shot, CoT), CMMLU (Li et al., 2023)(5-shot), SuperGPQA (Du et al., 2025)(5-shot), C-Eval (Huang et al., 2023)(5-shot), and BBH (Suzgun et al., 2022)(3-shot, CoT).
- **Math & STEM Tasks:** GSM8K (Cobbe et al., 2021)(4-shot, CoT), MATH (Hendrycks et al., 2021)(4-shot, CoT), and GPQA-Diamond (Rein et al., 2024)(5-shot).
- **Coding Tasks:** HumanEval+ (Liu et al., 2023)(0-shot), MBPP+ (Liu et al., 2023)(0-shot), MultiPL-E (Cassano et al., 2023)(0-shot)(Python, C++, Java, PHP, TypeScript, C#, Bash, JavaScript), and CRUX-O (Gu et al., 2024)(1-shot, CoT).

### B.2 Evaluation Benchmarks for Post-Training

To evaluate the quality of instruction-tuned models, we evaluate LLMs on a series of post-training benchmarks. The post-training benchmarks can be categorized into several dimensions:

- **General Tasks:** For general language understanding tasks, we utilize various benchmarks including MMLU (Hendrycks et al., 2020), MMLU-Pro (Wang et al., 2024a), CMMLU (Li et al., 2023), SuperGPQA (Du et al., 2025), BBH (Suzgun et al., 2022), DROP (Dua et al., 2019), C-Eval (Huang et al., 2023), and AGIEval (Zhong et al., 2023).

Table 6: Ablation studies on different routing strategies.

| | - | - | $g = 64$; $h = 128$ (Ours) | $g = 128$; $h = 128$ | $g = 64$; $h = 128$ |
|---|---|---|---|---|---|
| Architecture | MoE | MoE | Grove MoE | - | - |
| Routing Strategy | Single Router | Single Router | Single Router | Single Router | Coupled Router |
| # Total Params | 30B | 30B | 33B | 36B | 33B |
| # Activated Params | 3B | 3B | 3.14B-3.28B | 3.28B | 3.28B |
| # Avg. Activation | 3B | 3B | 3.26B | 3.28B | 3.28B |
| # Training Tokens | 0B | 50B | 50B | 50B | 50B |
| MMLU | 81.58 | 82.56 | 82.62 | 81.89 | 82.06 |
| CMMLU | 80.63 | 86.54 | 86.55 | 86.75 | 86.43 |
| SuperGPQA | 36.10 | 35.93 | 36.32 | 36.09 | 36.50 |
| GSM8K | 89.39 | 89.54 | 90.83 | 91.05 | 90.83 |
| MATH | 59.75 | 65.90 | 65.86 | 66.03 | 65.94 |
| GPQA-Diamond | 39.39 | 38.38 | 43.94 | 40.91 | 42.42 |
| HumanEval+ | 83.54 | 83.54 | 84.76 | 84.15 | 85.37 |
| MBPP+ | 71.96 | 74.34 | 75.13 | 75.40 | 75.13 |
| Average | 67.79 | 69.59 | 70.75 | 70.28 | 70.58 |

- **Alignment Tasks:** To evaluate how well the model aligns with human preferences, we employ a suite of specialized benchmarks. For instruction-following performance, we report the average prompt-level and instruction-level strict accuracy of IFEval (Zhou et al., 2023). To assess alignment with human preferences on general topics, we utilize Arena-Hard (Li et al., 2024).

- **Math & STEM Tasks:** For evaluating mathematical reasoning skills, we employ MATH (Hendrycks et al., 2021), MATH-500 (Lightman et al., 2023), Omni-MATH (Gao et al., 2024), AIME24 (AIME, 2025), and AIME25 (AIME, 2025). For STEM tasks, we utilize GPQA-Diamond (Rein et al., 2024) and OlympiadBench (He et al., 2024) as evaluation benchmarks. For AIME problems, we sample 16 times for each question and take the average accuracy as the final score. For GPQA-Diamond, we sample 8 times for each query and report the average accuracy.

- **Coding & Agent Tasks:** To test the LLM's proficiency in coding and agent-based tasks, we use HumanEval+ (Liu et al., 2023), MBPP+ (Liu et al., 2023), MultiPL-E (Cassano et al., 2023), LiveCodeBench (Jain et al., 2024) (v5, 2024.10-2025.02 and v6, 2025.02-2025.05), and BFCL v3 (Live) (Yan et al., 2024). For BFCL v3 (Live), all models are evaluated using the prompt format.

Notably, we configure the maximum output length to 16K to avoid overly lengthy output for non-reasoning LLMs during the evaluation process (ByteDance-Seed, 2025; Yang et al., 2025).

## C  ABLATION STUDIES

### C.1  ROUTING STRATEGY

To further investigate the effectiveness of our routing strategy, we perform a series of ablation studies. As presented in Table 6, our GroveMoE architecture with dynamic activation achieves performance comparable to architectures employing fixed expert activation. Specifically, the GroveMoE architecture maintains competitive performance while ensuring efficiency through adaptive computation. This finding demonstrates the efficacy and necessity of the adaptive routing mechanism in mitigating the performance disparity while effectively controlling computational cost. The competitive performance of GroveMoE, even when compared to an architecture that activates a fixed number of adjugate experts, is noteworthy. This outcome suggests that our GroveMoE successfully learns to discern token complexity, enabling dynamic computation.

### C.2  ENTIRE TRAINING FLOW

To ensure a fair comparison under the same training conditions, we conducted an additional ablation study, the results of which are presented in Table 7. We perform this study using the Qwen3-30B-A3B-Base model, undergoing mid-training for 50B tokens, and applying the same post-training

Table 7: Ablation studies on the entire training flow. The highest score is shown in bold.

| Models | Qwen3-30B-A3B | GroveMoE (Ours) |
|---|---|---|
| # Total Params | 30B | 33B |
| # Activated Params | 3B | 3.14B-3.28B |
| # Avg. Activation | 3B | 3.26B |
| # Training Tokens | 50B | 50B |
| # SFT Epochs | 1 | 1 |
| MMLU | 82.23 | **83.50** |
| MMLU-Pro | 67.21 | **68.91** |
| SuperGPQA | 42.02 | **45.39** |
| BBH | 84.14 | **85.15** |
| CEval | **84.23** | 83.93 |
| IFEval | 81.54 | **85.74** |
| GSM8K | 90.45 | **93.03** |
| MATH-500 | 86.40 | **88.30** |
| GPQA-Diamond | 56.19 | **58.21** |
| HumanEval+ | 85.06 | **85.67** |
| MBPP+ | 75.93 | **76.98** |
| MultiPLE | 68.27 | **69.47** |
| LiveCodeBench v5 | 26.80 | **27.54** |
| Average | 71.57 | **72.45** |

setup for one epoch. As shown in Table 7, our GroveMoE consistently outperforms the Qwen3-30B-A3B baseline across most metrics, which further affirms its architectural benefits.

## D  GROUPING STRATEGY

The expert grouping strategy represents a fascinating and potentially important research avenue. As detailed in Equation (2) and Equation (3), the grouping mechanism within our GroveMoE architecture utilizes a simple partition. Although a grouping strategy predicated on co-activation patterns or functional similarity could yield substantial benefits, we deliberately avoided an in-depth, data-driven analysis during training. This choice mitigates potential methodological issues, such as introducing bias or data leakage when using a validation set to determine the grouping. The significant gains in performance and efficiency achieved, despite employing this relatively simple grouping method, underscore the robustness of the core GroveMoE architecture.

## E  DEPLOYMENT OF GROVEMOE-INST

In our native PyTorch implementation, the Grove MoE architecture achieves the claimed theoretical speed. However, in our SGLang (sgl-project, 2025) implementation, the inference speed of GroveMoE-Inst is approximately 30% slower than the Qwen3-30B-A3B (Yang et al., 2025) baseline, which is an overhead that significantly exceeds the theoretical maximum 10% increase in activated parameters. This discrepancy arises because our implementation uses the generic MoE kernel (namely fused-moe) of SGLang for accessibility, which requires two separate kernel calls per GroveMoE layer. A customized and unified kernel designed to process both expert types in one operation would mitigate this latency and align performance more closely with theoretical expectations. The development of such a kernel is our key priority for future efficiency improvements.

## LIMITATIONS

Although our Grove MoE architecture provides a solid foundation, two primary limitations constrain its current potential and guide our future research. The first limitation stems from a scarcity of long-CoT data within the mid-training corpus. This data deficiency curtails the model's capacity for advanced reasoning, creating a capability gap compared to instruction-tuned LLMs that possess stronger foundational reasoning abilities, such as Qwen3-30B-A3B-2507 (Yang et al., 2025), etc. The second limitation is the exclusive reliance on rejection sampling for model refinement, without the integration of RL techniques. While rejection sampling has been effective, we anticipate that

incorporating RL methods will significantly enhance the model's overall capabilities. This remains a key objective for future development.

