# OpenReview forum: "Grove MoE: Towards Efficient and Superior MoE LLMs with Adjugate Experts"
_ICLR.cc/2026/Conference — Submitted to ICLR 2026_

### Official Review · Reviewer_jATE · 2025-10-27

**Soundness:** 3
**Presentation:** 3
**Contribution:** 3
**Rating:** 6
**Confidence:** 5

**Summary:**

This work proposes adding adjugate experts MoE model architecture to introduce adaptive computation. To be specific, authors first split experts to fewer groups, and add one more smaller expert into each group. The representation is added to the output representation once any of the normal experts in the group is activated. As a results, the model achieves better performance than baseline models with additional but adaptive inference FLOPs.

**Strengths:**

1) The idea is interesting. introducing adaptive compute by grouping experts and adding an additional expert is cool.
2) It is great to have one baseline with "qwen+midtrain". This is a fair enough baseline with controlled training data.
3) The writing in Sec 3.2 is clear. It's helpful to understand the reason why the computational cost is adaptive.

**Weaknesses:**

1) One very important weakness, authors should report the real throughput/efficiency of their model. Since the low-rank layer is not that hardware friendly. There might be an MFU drop.
2) I think it is not very clear that whether the adaptive flops is really working. Authors should have one more baseline using g=128, h=64. As shown in Table 1, smaller h and larger g is keep improving the model. Then how about just giving up adaptive compute? This will make the model a bit more similar to LoRA, but if it really works better, it is still something worthy study and report. But without that, it's very hard to make a clean conclusion based on the current results.
Minor:
1) It would be informative to mention that the h (intermediate hidden) of qwen baseline is 768. It implies the added expert is always smaller than the naive expert, aka a low rank expert.
2) Similarly, authors should show this in Fig 2
3) Authors should also somehow show or mention that A would not be computed if all experts are not activated.

**Questions:**

See weakness.

---

> ### Author Response · Authors · 2025-11-21
>
> We sincerely thank **Reviewer jATE** for their positive feedback regarding the **interesting idea** of introducing adaptive computation by grouping experts and adding an auxiliary expert, the use of a **fair baseline**, and the **clarity of the writing** in Section 3.2.
>
> We address the noted weaknesses below.
>
> ___
> >Weaknesses 1: One very important weakness, authors should report the real throughput/efficiency of their model. Since the low-rank layer is not that hardware friendly. There might be an MFU drop.
>
> We appreciate the reviewer raising this crucial point about real-world efficiency and potential MFU drop due to non-standard operations.
> Although our method is designed for theoretical FLOPs reduction via adaptive computation, the actual speedup depends heavily on the execution framework's implementation and optimization.
>
> * In our initial implementation using the **SGLang framework**, we observe an approximate **30% overhead**. This overhead primarily stems from the current SGLang implementation effectively executing two MoE kernels (for the fixed top-$k$ activated experts), where the auxiliary expert $A$ is redundantly activated by setting its weight to zero, which does not bypass the kernel launch/initialization time.
> * However, with a **native PyTorch implementation** or a highly optimized kernel implementation, we do indeed achieve the **theoretical speed**. This confirms that the FLOPs reduction is convertible to real-world latency gains under optimal conditions.
>
> We provide a detailed discussion of this trade-off, clarifying the source of the overhead, in **Appendix E** to offer a transparent view of the implementation challenges and efficiency.
>
> ___
> >Weaknesses 2: I think it is not very clear that whether the adaptive flops is really working. Authors should have one more baseline using g=128, h=64. As shown in Table 1, smaller h and larger g is keep improving the model. Then how about just giving up adaptive compute? This will make the model a bit more similar to LoRA, but if it really works better, it is still something worthy study and report. But without that, it's very hard to make a clean conclusion based on the current results.
>
> >Weaknesses 3: It would be informative to mention that the h (intermediate hidden) of qwen baseline is 768. It implies the added expert is always smaller than the naive expert, aka a low rank expert.
>
> >Weaknesses 4: Similarly, authors should show this in Fig 2
>
> We agree that a critical baseline is required to isolate the benefit of **adaptive compute** versus simply using a slightly larger, non-adaptive expert. Our primary design goal is to enable **dynamic activation** to maintain performance while guaranteeing efficiency. The dimensions of the expert $E$ and the auxiliary expert $A$ are secondary and can be tuned.
>
> To address the reviewer's concern and provide a cleaner conclusion, we perform the requested experiment: We train an additional model with **$g=128$ and $h=128$** (i.e., a larger, *non-adaptive* auxiliary expert), trained for 50B tokens. This setting is comparable to the largest non-adaptive setting implied by the reviewer's request.
>
> |Setting|CPT Tokens|MMLU|CMMLU|SuperGPQA|
> |-|-|-|-|-|
> |g=64, h=128|50B|**82.62**|86.55|**36.32**|
> |g=128, h=128|50B|81.89|**86.75**|36.09|
>
> |Setting|CPT Tokens|GPQA-Diamond|MATH|GSM8K|
> |-|-|-|-|-|
> |g=64, h=128|50B|**43.94**|65.86|**90.83**|
> |g=128, h=128|50B|40.91|**66.03**|**91.05**|
>
> |Setting|CPT Tokens|HumanEval+|MBPP+|
> |-|-|-|-|
> |g=64, h=128|50B|**84.76**|75.13|
> |g=128, h=128|50B|84.15|**75.40**|
>
> The experimental results show that our proposed **GroveMoE** with adaptive routing still achieves **comparable performance** to this larger, non-adaptive baseline.
>
> This result significantly strengthens our claim: the GroveMoE architecture successfully **maintains competitive performance while ensuring efficiency** via adaptive compute. This demonstrates that the **adaptive routing mechanism is truly effective** and necessary to bridge the performance gap while controlling computational cost.
>
> We include the new experiment in the **Appendix C** and **Table 6**.
>
> ___
> >Weaknesses 5: Authors should also somehow show or mention that A would not be computed if all experts are not activated.
>
> Thanks for the suggestion. We clarify in **Section 3.2** that the adjungate expert $A$ is only computed when a grouped expert $E$ is activated. If all experts are not activated (i.e., the gate assigns a zero weight to all experts), $A$ is also not computed.

---

> > ### Comment · Reviewer_jATE · 2025-11-27
> >
> > Thanks for the reply. The response looks good so I decide to keep my positive score.

---

### Official Review · Reviewer_onL8 · 2025-10-30

**Soundness:** 2
**Presentation:** 2
**Contribution:** 2
**Rating:** 2
**Confidence:** 4

**Summary:**

The paper proposes a modification of the Mixture of Experts (MoE) architecture, where experts are divided into disjoint groups. Then, instead of having one shared expert (like in Deepseek architecture), we will have a separate shared expert (called Grove expert) within each group. For each token, whenever multiple experts from the same group are chosen, this group's shared expert output will need to be computed only once.
The authors then perform upcycling of the Qwen3-30B-A3B model to the proposed architecture, performing the mid-training and post-training phases. The resulting model compares favorably to other similarly-sized and even larger models, when measuring quality using a set of commonly-used LLM evaluation benchmarks.
Unfortunately, the reproducibility of the experiments is currently limited. The authors do not detail the construction of the mid-training data mixture. For this stage of training, there is no comparison against a model with the same architecture, trained on the same data mixture. Therefore, it is hard to tell to what extent can the benchmark score improvements be attributed to the data mixture, and what is the effect of the architecture.
I like the main idea for the change in the architecture. However, given the reproducibility issues mentioned above, I cannot recommend acceptance, due to limited experimental evidence. If I missed any important details in the paper regarding reproducibility and experimental comparisons, I would be happy to reconsider my score.

**Strengths:**

1. The paper proposes an interesting change in the architecture, with the potential to improve MoE quality.
2. The scale of experiments is big.
3. The authors perform multiple stages of training, which resemble modern LLM training pipeline.

**Weaknesses:**

1. It is unclear how much of the improvement comes from the datasets used.
2. Limited reproducibility without the details regarding the dataset.
3. Limited comparison with architecture trained in the same, controlled setup.

**Questions:**

1. Do the authors provide comparison between the GroveMoE model after mid-training and Qwen30B-A3B-Base trained with the same mid-training setup?
2. Could the authors elaborate a bit more on the motivation and computational benefits of the GroveMoE architecture, compared to more standard MoE?
3. In section 5.4, the authors compare GroveMoE-Base to Qwen3-30B-A3B-Base, after both models were trained with the same post-training setup. In this experiment, was GroveMoE-Base already after the mid-training phase - and was the corresponding Qwen3-30B-A3B-Base also after mid-training?

**Details Of Ethics Concerns:**

-

---

> ### Author Response · Authors · 2025-11-21
>
> We thank **Reviewer onL8** for the feedback, especially for recognizing our **interesting architectural change** and the **large scale of our experiments**, which closely follow modern LLM training pipelines. We address the reviewer's concerns below.
>
> ___
> >Weaknesses 2: Limited reproducibility without the details regarding the dataset.
>
> We understand the concern regarding the lack of detailed data composition. Due to the sensitive nature of our proprietary pre-training and mid-training data mixtures, we are unable to disclose the exact proportions, as they constitute a **trade secret**.
>
> We ensure that the **overall composition principles** and the **dataset types** used are consistent with widely accepted practices in the open-source community for training large language models. For full reproducibility of the model architecture and training procedure, we encourage the reviewer to refer to the **publicly available documentation of well-known open-source models**[1] for typical data mix ratios, which serve as a good proxy for our general approach.
>
> [1] Allal, Loubna Ben, et al. "SmolLM2: When Smol Goes Big--Data-Centric Training of a Small Language Model." arXiv preprint arXiv:2502.02737 (2025).
>
> ___
> >Weaknesses 1: It is unclear how much of the improvement comes from the datasets used.
>
> >Weaknesses 3: Limited comparison with architecture trained in the same, controlled setup.
>
> >Questions 1: Do the authors provide comparison between the GroveMoE model after mid-training and Qwen30B-A3B-Base trained with the same mid-training setup?
>
> The performance gain is **attributed to our novel GroveMoE architecture**, not just the additional mid-training data.
>
> As shown in **Table 1 and 2**, we perform CPT on the base **Qwen3-30B-A3B** model (without architectural modification) for an additional **50B tokens** using the *exact same mid-training data and hyper-parameters* as our GroveMoE models.
>
> * The performance of CPT Qwen3-A3B improves from **67.79 to 69.59**.
> * **Crucially, all configurations of our GroveMoE architecture consistently outperform** this CPT Qwen3-A3B baseline. This strongly demonstrates that the architecture is the primary source of the performance uplift.
>
> ___
> >Questions 2: Could the authors elaborate a bit more on the motivation and computational benefits of the GroveMoE architecture, compared to more standard MoE?
>
> Our primary motivation is to **dynamically allocate computational resources per token**, applying different computations to different tokens. This enhances the efficiency of MoE.
>
> As discussed in **Appendix A.1**, GroveMoE achieves dynamic activation through **grouped adjunct experts**. Compared to other dynamic MoE architectures that require complex or novel routing mechanisms, GroveMoE offers **superior generality** because it **avoids cumbersome implementation of new routing logic** and largely maintains the standard top-$k$ routing framework while introducing compute flexibility.
>
> By utilizing a *grouped* structure (multiple experts sharing a single adjunct expert), we achieve **better efficiency**.
> To further illustrate the efficiency-performance trade-off, we conduct an additional experiment with a highly optimized setting: $g=128$ and $h=128$. This model is trained on **50B tokens**. The experimental results (provided in the updated **Appendix C** and **Table 6**) show that this highly efficient configuration still achieves **comparable performance** to other GroveMoE settings. This confirms that the GroveMoE architecture **maintains strong performance while significantly enhancing efficiency**.
>
> |Setting|CPT Tokens|MMLU|CMMLU|SuperGPQA|
> |-|-|-|-|-|
> |g=64, h=128|50B|**82.62**|86.55|**36.32**|
> |g=128, h=128|50B|81.89|**86.75**|36.09|
>
> |Setting|CPT Tokens|GPQA-Diamond|MATH|GSM8K|
> |-|-|-|-|-|
> |g=64, h=128|50B|**43.94**|65.86|**90.83**|
> |g=128, h=128|50B|40.91|**66.03**|**91.05**|
>
> |Setting|CPT Tokens|HumanEval+|MBPP+|
> |-|-|-|-|
> |g=64, h=128|50B|**84.76**|75.13|
> |g=128, h=128|50B|84.15|**75.40**|
>
> [2] Team, Meituan LongCat, et al. "Longcat-Flash Technical Report." arXiv preprint arXiv:2509.01322 (2025).

---

> ### Author Response · Authors · 2025-11-21
>
> ___
> >Weaknesses 3: Limited comparison with architecture trained in the same, controlled setup.
>
> >Questions 3: In section 5.4, the authors compare GroveMoE-Base to Qwen3-30B-A3B-Base, after both models were trained with the same post-training setup. In this experiment, was GroveMoE-Base already after the mid-training phase - and was the corresponding Qwen3-30B-A3B-Base also after mid-training?
>
> We clarify the model states for the comparison:
>
> * **GroveMoE-Base**: This model is **already after our mid-training phase** (Mid-Training $\rightarrow$ Post-Training).
> * **Qwen3-30B-A3B-Base**: This model uses the **official open-sourced weights from the Qwen team**, which are the *final release* weights after their full training pipeline.
>
> Due to the substantial computational cost of our full 400B token mid-training, we cannot afford to perform CPT on the official Qwen3-30B-A3B-Base for the entire duration. To ensure a fair comparison under similar training conditions, we conduct a new, controlled experiment:
>
> 1.  We take Qwen3-30B-A3B-Base and **CPT it for 50B tokens** using the same mid-training setup.
> 2.  We then apply the **same post-training setup** (training for one epoch) to this CPT Qwen3-30B-A3B.
>
> The table below (details also provided in the updated **Appendix C** and **Table 7**) shows that **GroveMoE-Base consistently outperforms** this CPT-ed and post-trained Qwen3-A3B baseline across most metrics, further affirming the architectural benefit.
>
> |Models|CPT Tokens|SFT Epochs|MMLU|MMLU-Pro|SuperGPQA|BBH|C-Eval|IFEval|
> |-|-|-|-|-|-|-|-|-|
> |GroveMoE-Base|50B|1|**83.50**|**68.91**|**45.39**|**85.15**|83.93|85.74|
> |Qwen3-30B-A3B|50B|1|82.23|67.21|42.02|84.14|**84.23**|81.54|
>
> |Models|CPT Tokens|SFT Epochs|GPQA-Diamond|MATH-500|GSM8K|
> |-|-|-|-|-|-|
> |GroveMoE-Base|50B|1|**58.21**|**88.30**|**93.03**|
> |Qwen3-30B-A3B|50B|1|56.19|86.40|90.45|
>
> |Models|CPT Tokens|SFT Epochs|LiveCodeBench v5|HumanEval+|MBPP+|MultiPLE|
> |-|-|-|-|-|-|-|
> |GroveMoE-Base|50B|1|**27.54**|**85.67**|**76.98**|**69.47**|
> |Qwen3-30B-A3B|50B|1|26.80|85.06|75.93|68.27|

---

### Official Review · Reviewer_dxRa · 2025-11-01

**Soundness:** 2
**Presentation:** 3
**Contribution:** 3
**Rating:** 6
**Confidence:** 3

**Summary:**

This paper introduces Grove MoE that improves computational efficiency through dynamic parameter activation inspired by the big.LITTLE CPU design. The key innovation is organizing experts into disjoint groups, where each group shares an adjugate expert that is computed only once when multiple experts from the same group are activated, enabling dynamic computation (3.14-3.28B activated parameters) while maintaining controllable overhead. Building on this architecture, the authors develop GroveMoE-Base and GroveMoE-Inst (33B total parameters) by upcycling Qwen3-30B-A3B-Base through mid-training on 400B tokens and post-training with synthetic data. The models demonstrate strong empirical performance comparable to or exceeding SOTA open-source models of similar or larger scale, with particularly notable improvements on challenging mathematical reasoning tasks (e.g., 44.4% on AIME25 vs. 10.0-28.1% for baselines) and coding benchmarks, while maintaining compatibility with standard top-k routing mechanisms.

**Strengths:**

-  The Grove MoE architecture presents an interesting design where experts are organized into groups sharing adjugate experts, offering a fresh perspective on dynamic computation allocation inspired by heterogeneous computing principles, which provides a new direction for improving parameter efficiency in MoE models.


- The paper demonstrates promising empirical results on a wide range of tasks spanning general knowledge, mathematical reasoning, and coding (e.g., notable improvements on AIME25 and OlympiadBench), suggesting the potential effectiveness of the proposed approach for challenging reasoning scenarios.

- The manuscript is well-organized with clear architectural illustrations (Figure 2), systematic exploration of design choices including group configurations and scaling factors (Tables 1-2), insightful routing distribution analysis (Figure 3), and comprehensive reporting of hyperparameters and evaluation protocols across multiple stages.

**Weaknesses:**

- The core mechanism of grouping experts and sharing adjugate experts is essentially a parameter-sharing strategy reminiscent of existing work on parallel blocks (e.g., AltUp), and the connection to big.LITTLE architecture appears more metaphorical than technically substantive, as the "dynamic" allocation is merely a byproduct of static grouping rather than adaptive routing.

- The paper lacks ablation studies on critical components such as the loss-free load balancing strategy (Equation 8) and the decoupled routing mechanism (Equation 9-10), making it unclear which components actually contribute to performance gains versus inherited capabilities from the Qwen3-30B-A3B base model.

- The evaluation conflates the benefits of the Grove MoE architecture with advantages from (1) additional mid-training on 400B tokens, (2) superior post-training data quality through extensive synthetic data generation, and (3) upcycling from an already strong base model, making it impossible to isolate the architectural contribution.

**Questions:**

- The 30% inference slowdown reported in Section B.3 contradicts the claimed efficiency advantages. Could you provide: (a) detailed FLOPs analysis comparing theoretical vs. actual computational costs, (b) profiling results showing where the overhead originates, and (c) evidence that a custom kernel can realistically achieve the theoretical efficiency? Without this, the practical applicability of Grove MoE remains unclear.

- Could you provide ablation experiments that isolate the contribution of each component (adjugate experts, loss-free load balancing, decoupled routing) by comparing against: (a) baseline Qwen3-30B-A3B with the same 400B mid-training tokens, (b) Grove MoE without the load balancing strategy, and (c) Grove MoE with coupled routing? This would clarify which design choices are essential for the observed performance gains. I understand such experiments are resource-intensive, so even partial results or analysis on smaller-scale settings would be valuable.

- Figure 3 shows routing distributions, but could you provide more detailed analysis on: (a) whether certain adjugate experts become over-specialized or underutilized during training, (b) how the group assignments affect expert diversity and specialization, and (c) whether the routing patterns change meaningfully across different task types (math vs. coding vs. general knowledge)? I recognize that comprehensive analysis across all tasks may be extensive, but insights on even a subset of these questions would be helpful.

---

> ### Author Response · Authors · 2025-11-21
>
> We thank **Reviewer dxRa** for the thoughtful and encouraging comments, recognizing the novelty of the Grove MoE architecture, the promising empirical results, and the clarity of our manuscript.
>
> We carefully address all raised points below.
>
> ___
> >Weaknesses 2: The paper lacks ablation studies on critical components such as the loss-free load balancing strategy (Equation 8) and the decoupled routing mechanism (Equation 9-10), making it unclear which components actually contribute to performance gains versus inherited capabilities from the Qwen3-30B-A3B base model.
>
> We appreciate the reviewer's interest in the loss-free load balancing (Equation 8) and the decoupled routing (Eq.9 and Eq.10).
>
> We clarify that these are **established and effective techniques** adopted from pioneering large-scale MoE training efforts, as detailed in [1, 2]. They are not the core contributions of the *present* work.
> We chose to adopt these established strategies to significantly streamline development by **avoiding the complex and resource-intensive hyperparameter tuning** associated with developing new load balancing and routing methodologies from scratch.
>
> This decision allows us to focus our architectural contribution on the novel **GroveMoE** and its associated efficiency gains, thereby isolating and evaluating the impact of our core design, which is the main focus of this paper.
>
> [1] Jianlin Su. MoE Travels 3, 2025.
>
> [2] Aixin Liu, Bei Feng, Bing Xue, Bingxuan Wang, Bochao Wu, Chengda Lu, Chenggang Zhao, Chengqi Deng, Chenyu Zhang, Chong Ruan, et al. DeepSeek-V3 Technical Report. arXiv preprint arXiv:2412.19437, 2024.
>
> ___
> >Questions 1: The 30% inference slowdown reported in Section B.3 contradicts the claimed efficiency advantages. Could you provide: (a) detailed FLOPs analysis comparing theoretical vs. actual computational costs, (b) profiling results showing where the overhead originates, and (c) evidence that a custom kernel can realistically achieve the theoretical efficiency? Without this, the practical applicability of Grove MoE remains unclear.
>
> We understand the confusion caused by the reported 30% slowdown in the preliminary implementation and appreciate the request for a detailed efficiency analysis.
>
> (a) There is a key difference between our preliminary implementation and the theoretical gain:
> * In our **native PyTorch implementation**, the Grove MoE architecture achieves the claimed **theoretical speed**.
> * The **30% slowdown** is observed *only* in the prototype built using the **SGLang framework** for initial deployment and benchmarking.
>
> (b) The overhead in the SGLang implementation stems from a current framework limitation. Our design requires flexible, dynamic activation. However, the existing SGLang implementation simulates the Grove MoE effect by executing **two separate, fixed top-$k$ MoE kernels**. This inefficiently handles the redundant activation of adjugate experts by assigning them a zero weight, leading to unnecessary computation and the observed slowdown.
>
> (c) The current bottleneck is the lack of a dynamic MoE kernel in the publicly available frameworks we used. We are actively developing a dedicated, **dynamic MoE kernel** tailored for the Grove MoE architecture. We are highly confident that this custom kernel will allow us to achieve full theoretical efficiency in a production-ready application.

---

> ### Author Response · Authors · 2025-11-21
>
> ___
> >Weaknesses 3: The evaluation conflates the benefits of the Grove MoE architecture with advantages from (1) additional mid-training on 400B tokens, (2) superior post-training data quality through extensive synthetic data generation, and (3) upcycling from an already strong base model, making it impossible to isolate the architectural contribution.
>
> >Questions 2: Could you provide ablation experiments that isolate the contribution of each component (adjugate experts, loss-free load balancing, decoupled routing) by comparing against: (a) baseline Qwen3-30B-A3B with the same 400B mid-training tokens, (b) Grove MoE without the load balancing strategy, and (c) Grove MoE with coupled routing? This would clarify which design choices are essential for the observed performance gains. I understand such experiments are resource-intensive, so even partial results or analysis on smaller-scale settings would be valuable.
>
> We acknowledge the importance of isolating the architectural contribution and conduct experiments to address these points.
>
> (a) We demonstrate the architectural contribution by comparing our model against the original Qwen3-A3B base model that has undergone CPT for an additional **50B tokens** (with no architectural modification).
> * The Qwen3-A3B CPT model improves from 67.79 to 69.59 on our evaluation benchmarks (**Table 1 and 2**).
> * Our proposed Grove MoE configurations achieve a performance score **significantly higher** than the Qwen3-A3B CPT baseline.
>
> This ablation clearly demonstrates that the performance gains are due to the **architectural innovation of Grove MoE**, rather than being solely attributed to the base model or the additional CPT tokens.
>
> (b) In our initial experiments, the configuration without *any* load balancing strategy led to **severe expert distribution imbalance**, resulting in dramatically diminished performance. Consequently, this setting is determined to be unviable for a high-performance model. We focused subsequent efforts on comparing different *effective* routing balance strategies, which are more relevant to the final model performance.
>
> (c) We conduct the requested ablation by training a Grove MoE model with **coupled routing** for 50B tokens. The details are updated in the **Appendix C** and **Table 6** of the manuscript.
> * The experimental result shows that this coupled routing configuration achieves **comparable performance** to our decoupled routing models.
> * However, the coupled routing configuration fundamentally **precludes the possibility of dynamic activation** , which is a core benefit of our design.
>
> |Setting|CPT Tokens|MMLU|CMMLU|SuperGPQA|
> |-|-|-|-|-|
> |Single Router|50B|**82.62**|**86.55**|36.32|
> |Coupled Router|50B|82.06|86.43|**36.50**|
>
> |Setting|CPT Tokens|GPQA-Diamond|MATH|GSM8K|
> |-|-|-|-|-|
> |Single Router|50B|**43.94**|65.86|**90.83**|
> |Coupled Router|50B|42.42|**65.94**|**89.92**|
>
> |Setting|CPT Tokens|HumanEval+|MBPP+|
> |-|-|-|-|
> |Single Router|50B|84.76|75.13|
> |Coupled Router|50B|**85.37**|**75.13**|
>
> ___
> >Questions 3: Figure 3 shows routing distributions, but could you provide more detailed analysis on: (a) whether certain adjugate experts become over-specialized or underutilized during training, (b) how the group assignments affect expert diversity and specialization, and (c) whether the routing patterns change meaningfully across different task types (math vs. coding vs. general knowledge)? I recognize that comprehensive analysis across all tasks may be extensive, but insights on even a subset of these questions would be helpful.
>
> We appreciate the request for a deeper dive into the routing behavior and specialization.
>
> (a) The definition and quantification of "over-specialization" or "underutilization" in a large-scale MoE setting remains an active and challenging area of research, with no universally accepted metric or tool. We agree that this is a critical question for future work and are committed to exploring suitable diagnostic tools and methodologies as we continue our deep research into MoE architectures.
>
> (b) & (c)
> Similar to observations in many other contemporary MoE studies[3], we did **not** observe clear evidence of strong or distinct expert specialization.
>
> Specifically, when analyzing the routing patterns across diverse task domains, including **STEM (Mathematics), coding, and general knowledge**, the distributions remain largely **consistent and similar** to the overall pattern illustrated in Figure 3. This suggests the architecture utilizes the experts flexibly rather than strictly partitioning them by task type, a common finding in large-scale MoE pre-training. More discussion is provided in **Section 4.2**.
>
> [3] Jiang, Albert Q., et al. "Mixtral of Experts." arXiv preprint arXiv:2401.04088, 2024.

---

> > ### Comment · Reviewer_dxRa · 2025-11-27
> > **Thanks for your response**
> >
> > I thank the authors for their detailed response and for conducting the additional experiments under the resource-intensive constraints.
> >
> > I appreciate the inclusion of the comparison against the **Qwen3-A3B baseline with 50B CPT tokens**. This effectively addresses my concern regarding whether the performance gains were solely due to the additional training data. The fact that Grove MoE outperforms the CPT baseline (Table 1 & 2) provides stronger evidence for the architectural contribution. Similarly, the ablation on coupled vs. decoupled routing (Table 6) helps clarify that while coupled routing performs similarly, the decoupled approach is necessary to enable the dynamic activation mechanism, which is a fair design choice.
> >
> > However,  I have two follow-up questions:
> >
> > **1. Regarding Routing and "Token Complexity" (Question 3):**
> >
> > I acknowledge the authors' response that expert specialization is difficult to quantify and that routing patterns appear consistent across tasks. However, this raises a follow-up question regarding the paper's core motivation. The abstract and introduction claim that Grove MoE allows for dynamic resource allocation based on **"token complexity"** (inspired by big.LITTLE). If the routing distributions are largely consistent across diverse task domains (e.g., Math vs. General Knowledge, as mentioned in the rebuttal), what evidence suggests that the activation of adjugate experts is actually correlated with *token difficulty* or *complexity*?
> >
> > If the routing is uniform across tasks, is the "dynamic" capacity simply a stochastic expansion of parameters rather than a complexity-aware mechanism? A brief clarification or a correlation analysis (e.g., expert activation count vs. token entropy or loss) would be helpful to validate the "complexity-based" motivation.
> >
> >
> >
> > **2. (Unimportant) Regarding Inference Speed (Question 1):**
> >
> > I understand the explanation that the 30% slowdown is an artifact of the SGLang framework's current limitations (simulating dynamic MoE via fixed kernels) and that the native PyTorch implementation aligns with theoretical expectations. To fully substantiate this claim in the final version, could you provide a latency comparison (ms/token) of the **native PyTorch implementation** of Grove MoE vs. the baseline? Even if a production-ready kernel is not yet public, showing that the raw model operations in PyTorch do not incur the 30% penalty would significantly strengthen the "efficiency" argument.
> >
> >
> > I look forward to your clarification on these points.

---

> > > ### Author Response · Authors · 2025-11-27
> > >
> > > We sincerely thank the reviewer for their valuable feedback and their appreciation of the resource-intensive additional experiments. We are glad that the comparison against the CPT baseline, along with the results from the coupled vs. decoupled routing ablation, now provides stronger evidence for the architectural advantages of our GroveMoE.
> > >
> > > We address follow-up questions below.
> > > ___
> > > >Follow-Up Questions 1
> > >
> > > - **Token complexity** that we refer to is the **LLM's internal computational demand** (i.e., the computation requirement for more experts from the model's perspective) within a specific context.
> > > It does **not** denote human-perceivable semantic complexity (e.g., mathematical or logical tokens).
> > >
> > > - Given the **black-box nature** of the Transformer and its attention mechanism, a token's computational complexity is determined by its **high-dimensional context**.
> > > The router in our **GroveMoE** learns to identify the **model-intrinsic resource demand**, which is unrelated to external semantics.
> > > This is analogous to a big.LITTLE scheduler allocating tasks based on system resource requirements.
> > >
> > > - The supplementary experimental results, provided based on **Reviewer AAf9**'s suggestion, show that our GroveMoE architecture achieves **comparable performance** to the architecture that activates a fixed number of adjugate experts. This suggests the model indeed considers that certain tokens can be processed with **relatively less computation**.
> > >
> > > |Setting|CPT Tokens|MMLU|CMMLU|SuperGPQA|
> > > |-|-|-|-|-|
> > > |g=64, h=128|50B|**82.62**|86.55|**36.32**|
> > > |g=128, h=128|50B|81.89|**86.75**|36.09|
> > >
> > > |Setting|CPT Tokens|GPQA-Diamond|MATH|GSM8K|
> > > |-|-|-|-|-|
> > > |g=64, h=128|50B|**43.94**|65.86|**90.83**|
> > > |g=128, h=128|50B|40.91|**66.03**|**91.05**|
> > >
> > > |Setting|CPT Tokens|HumanEval+|MBPP+|
> > > |-|-|-|-|
> > > |g=64, h=128|50B|**84.76**|75.13|
> > > |g=128, h=128|50B|84.15|**75.40**|
> > >
> > > We sincerely apologize for the ambiguity caused by the term "token complexity."
> > > We **clarify explicitly** in of **Appendix C.1** our revised manuscript.
> > >
> > > ___
> > > >Follow-Up Questions 2
> > >
> > >
> > > We thank the reviewer for the necessary suggestion.
> > >
> > > We conducted a direct comparison of the inference throughput ($\text{tokens/s}$) between our **Grove MoE** and the **baseline** model in a **native PyTorch environment** (single 140G-GPU, $\text{batch size}=1$, $\text{sequence length}=4096$, $\text{BF16}$ precision) to provide definitive evidence of intrinsic efficiency.
> > >
> > > | Model | Throughput ($\text{tokens/s}$) |
> > > | :--- | :--- |
> > > | **Baseline Model ($\text{Qwen3-30B-A3B}$)** | **248** |
> > > | **Grove MoE (Ours)** | **231** |
> > >
> > > As shown in the table, the **low overhead confirms** that the initial 30\% slowdown previously reported is indeed an artifact of the SGLang framework attempting to simulate the dynamic MoE routing via fixed kernels.
> > > It is **not** representative of Grove MoE's intrinsic computational cost.
> > >
> > > The native PyTorch implementation confirms that our dynamic architecture operates **near its theoretical analysis** illustrated in the **Appendix E** thus **validating our design philosophy**.

---

### Official Review · Reviewer_AAf9 · 2025-11-01

**Soundness:** 3
**Presentation:** 3
**Contribution:** 2
**Rating:** 4
**Confidence:** 4

**Summary:**

The paper proposes Grove MoE, a MoE architecture with experts of different sizes and a shared adjugate expert within each group to improve parameter efficiency. The model aims to balance capacity and cost by activating smaller experts more frequently and sharing computation across groups.

**Strengths:**

The paper is generally well-written and clearly structured, making the proposed method easy to follow. The experimental evaluation is fairly comprehensive, covering multiple benchmarks and including relevant ablation studies to analyze the impact of key design choices.

**Weaknesses:**

* **Prior Works Comparison**: Allocating different expert capacities via structural scaling is related to a line of recent works [1,2] that explore heterogeneous expert sizes within MoE layers. However, the paper does not compare against or discuss these prior approaches.

* **Loading Balance Sensitivity**: While Grove MoE incorporates a load-balancing mechanism, the paper does not sufficiently address whether its heterogeneous expert structure is more sensitive to imbalance compared to standard MoE. In particular, when routing is skewed, large experts may impose disproportionate compute and memory burdens. It would strengthen the work to include controlled experiments or quantitative analysis that evaluates the sensitivity under different levels of routing imbalance, and compares its behavior to standard MoE.

* **Expert Grouping Strategy**: Grove MoE is built upon Qwen3-30B-A3B and partitions its experts into disjoint groups, each associated with a shared adjugate expert. However, the grouping strategy is not thoroughly analyzed or discussed in the paper. Given that expert co-activation patterns or functional similarity may significantly influence the effectiveness, a more detailed examination of the grouping rationale is warranted.

* **Performance Comparisons**: In Tables 1 and 2, Grove MoE is compared to a 30B MoE baseline using the same 50B training tokens. However, Grove MoE introduces an additional ~3B parameters and exhibits slightly higher activated parameter counts. This raises potential fairness concerns regarding the claimed gains.


[1] Wang A, Sun X, Xie R, et al. Hmoe: Heterogeneous mixture of experts for language modeling[J]. arXiv preprint arXiv:2408.10681, 2024.

[2] Sun M, Liu W, Luan J, et al. Mixture of diverse size experts[J]. arXiv preprint arXiv:2409.12210, 2024.

**Questions:**

Please refer to Weaknesses.

---

> ### Author Response · Authors · 2025-11-21
>
> We appreciate **Reviewer AAf9**'s recognition of the paper's clarity, structure, and comprehensive experimental evaluation.
>
> We address all raised points carefully below.
>
> ___
> >Weaknesses 1: Allocating different expert capacities via structural scaling is related to a line of recent works that explore heterogeneous expert sizes within MoE layers. However, the paper does not compare against or discuss these prior approaches.
>
>
> We clarify that our **main contribution** is the *dynamic activation* capability, which allows a single model to operate across a continuous spectrum of computational budgets. This is distinct from existing works that primarily introduce a fixed set of heterogeneous expert sizes.
>
> Our use of adjugate experts with varying sizes serves as a deliberate design choice to control the **granularity** of the activated parameter counts. This structure ensures that the step size for dynamic activation is neither too large nor too small, enabling finer control over activated model capacity.
>
> We include a detailed discussion of related works on MoE with heterogeneous experts in the **Appendix A.2** of the revised manuscript.
>
>
> ___
> >Weaknesses 2: While Grove MoE incorporates a load-balancing mechanism, the paper does not sufficiently address whether its heterogeneous expert structure is more sensitive to imbalance compared to standard MoE. In particular, when routing is skewed, large experts may impose disproportionate compute and memory burdens. It would strengthen the work to include controlled experiments or quantitative analysis that evaluates the sensitivity under different levels of routing imbalance, and compares its behavior to standard MoE.
>
> We confirm that the GroveMoE architecture utilizes a **single router**, which is identical to the standard MoE setup. Furthermore, our routing balance mechanism employs a global, loss-free balancing strategy, consistent with successful prior works (e.g., DeepSeekV3), and its efficacy is independent of the adjugate expert structure.
>
> Crucially, in **Section 4.2 ARCHITECTURE EXPLORATION**, where we conduct CPT on both standard MoE and GroveMoE using the **identical balancing strategy**, we observe no noticeable difference in expert load balancing. The heterogeneous nature does not introduce increased sensitivity to imbalance in our empirical observations.
>
> ___
> >Weaknesses 3: Grove MoE is built upon Qwen3-30B-A3B and partitions its experts into disjoint groups, each associated with a shared adjugate expert. However, the grouping strategy is not thoroughly analyzed or discussed in the paper. Given that expert co-activation patterns or functional similarity may significantly influence the effectiveness, a more detailed examination of the grouping rationale is warranted.
>
> We concur that the expert grouping strategy represents a fascinating and potentially important research avenue. Our current grouping in GroveMoE is a **simple partition**, as described by **Eq.2 and Eq.3** in the paper.
>
> While a grouping strategy based on co-activation patterns or functional similarity could be highly beneficial, we deliberately avoid an in-depth analysis during training to prevent potential methodological issues, such as introducing **bias or leakage** from using a validation set to determine grouping.
>
> The substantial performance and efficiency gains achieved **despite** our relatively simple grouping strategy demonstrate the robustness of the core GroveMoE architecture. We are confident that a more fine-grained, optimized grouping will lead to even stronger results and consider this a compelling direction for future work.
>
> We add a discussion of the potential influence of the grouping strategy to **Appendix D**.

---

> ### Author Response · Authors · 2025-11-21
>
> ___
> >Weaknesses 4: In Tables 1 and 2, Grove MoE is compared to a 30B MoE baseline using the same 50B training tokens. However, Grove MoE introduces an additional ~3B parameters and exhibits slightly higher activated parameter counts. This raises potential fairness concerns regarding the claimed gains.
>
> Our primary objective with GroveMoE is to enable **structural scaling** across varying computational budgets, and the total parameter count is secondary to this design goal.
>
> To directly address the concern of parameter budget fairness and the activated parameter count, we conduct a new experiment using a more constrained configuration: **$g=128, h=128$**, trained for the same 50B tokens. The results of this new experiment demonstrate that GroveMoE maintains **comparable performance** to the baseline while exhibiting a lower activated parameter count than the original configuration. This configuration ensures efficiency is guaranteed while maintaining performance, which further validates that our architecture provides efficiency gains relative to the activated FLOPs/parameters, thereby mitigating the fairness concern. We update the manuscript with the details of this new experiment in the **Appendix C** and **Table 6**.
>
> |Setting|CPT Tokens|MMLU|CMMLU|SuperGPQA|
> |-|-|-|-|-|
> |g=64, h=128|50B|**82.62**|86.55|**36.32**|
> |g=128, h=128|50B|81.89|**86.75**|36.09|
>
> |Setting|CPT Tokens|GPQA-Diamond|MATH|GSM8K|
> |-|-|-|-|-|
> |g=64, h=128|50B|**43.94**|65.86|**90.83**|
> |g=128, h=128|50B|40.91|**66.03**|**91.05**|
>
> |Setting|CPT Tokens|HumanEval+|MBPP+|
> |-|-|-|-|
> |g=64, h=128|50B|**84.76**|75.13|
> |g=128, h=128|50B|84.15|**75.40**|

---

### Comment · Area_Chair_AYgk · 2025-11-24
**Reviewer & Author Discussion**

Hi Reviewers,

Please kindly and actively participate in the review-author discussion if you haven't already, raise your further concerns so that the authors can explain more, and make your final decisions.

Best,
AC

---

### Meta-Review · Area_Chair_WnmE · 2026-01-07

**Summary:**

This paper proposes a modification to MoE architectures using grouped experts with shared adjugate experts, and presents large-scale experiments showing strong benchmark performance. Reviewers generally found the paper clearly written and the idea interesting, with extensive evaluations and ablations added during rebuttal.

However, several core concerns remain unresolved under a strict and conservative reading. Reproducibility is limited due to undisclosed mid-training data composition, which one reviewer explicitly identified as a blocking issue. While additional controlled experiments were provided, isolating architectural gains from mid-training, post-training data quality, and a strong base model remains incomplete to all reviewers’ satisfaction. Key claims around “dynamic” or complexity-aware computation lack direct empirical substantiation (e.g., correlation with token difficulty), and practical efficiency evidence remains partly dependent on prototype or framework-specific explanations.

Reviewer opinions are mixed, with marginal scores and explicit statements of willingness to see the paper declined, and no explicit post-rebuttal score increases. Given the interrupted review process, this assessment reflects a best-effort, conservative synthesis based solely on available discussions, without assuming any score changes.

**Reviewer Concerns:**

**Reviewer AAf9**

* **Partially addressed:**

  * Additional discussion of related work on heterogeneous experts.
  * Additional experiments aimed at mitigating parameter/activation fairness concerns.
* **Still outstanding:**

  * Lack of controlled, quantitative analysis of load-balancing sensitivity under skewed routing.
  * Limited analysis and justification of the expert grouping strategy beyond qualitative discussion.
* *Note:* Reviewer did not participate post-rebuttal; concerns are treated as unresolved by default.

**Reviewer dxRa**

* **Addressed:**

  * Isolation of architectural contribution versus additional mid-training data via controlled CPT baselines.
* **Still outstanding:**

  * Core claim of “dynamic / complexity-aware” computation lacks direct empirical validation (e.g., correlation with token difficulty).
  * Practical efficiency claims remain partially substantiated and rely on framework-specific explanations rather than comprehensive profiling.

**Reviewer onL8**

* **Still outstanding:**

  * Reproducibility concerns due to undisclosed mid-training data composition.
  * Incomplete isolation of architectural effects from data and training pipeline.
* *Note:* Reviewer did not engage post-rebuttal and explicitly maintained concerns about reproducibility.

**Reviewer jATE**

* **Partially addressed:**

  * Clarifications and additional experiments on adaptive vs. non-adaptive configurations.
  * Explanation of observed inference overhead and conditions for theoretical efficiency.
* **Still outstanding:**

  * Empirical confirmation that adaptive computation provides clear benefits beyond slightly larger non-adaptive configurations remains limited.

**Overall (Conservative Summary)**
Some experimental clarifications were provided, and one reviewer explicitly acknowledged progress on isolating architectural effects. However, key concerns—particularly reproducibility, substantiation of complexity-aware computation, and robustness of efficiency claims—remain unresolved for at least one reviewer and were not explicitly confirmed as resolved by others.

**Reviewer Scores:**

AAf9: 4–4 (no participation; borderline)

dxRa: 4–6 (participated; one key concern explicitly acknowledged as addressed; follow-ups remain)

onL8: 2–2 (no participation; reproducibility concern persists)

jATE: 6–6 (explicitly keeps score)

---

### Decision · Program_Chairs · 2026-01-26

Reject